# A Review of the Role of Imaging Modalities in the Evaluation of Viral Myocarditis with a Special Focus on COVID-19-Related Myocarditis

**DOI:** 10.3390/diagnostics12020549

**Published:** 2022-02-21

**Authors:** Adedayo Adeboye, Deya Alkhatib, Asra Butt, Neeraja Yedlapati, Nadish Garg

**Affiliations:** 1Department of Cardiology, University of Tennessee Health Science Center, Memphis, TN 38163, USA; aadeboye@uthsc.edu (A.A.); dalkhati@uthsc.edu (D.A.); nyedlapa@uthsc.edu (N.Y.); 2Department of Cardiology, Veterans Affairs Medical Center, Memphis, TN 38104, USA; 3Department of Internal Medicine, University of Tennessee Health Science Center, Memphis, TN 38163, USA; akhalid2@uthsc.edu; 4Department of Cardiology, Methodist University Hospital, Memphis, TN 38104, USA; 5Department of Cardiology, Memorial Hermann Southeast Hospital, Houston, TX 77089, USA

**Keywords:** viral myocarditis, COVID-19-related myocarditis, echocardiography, cardiac magnetic resonance imaging (CMR), cardiac CT, PET-CT, SPECT

## Abstract

Viral myocarditis is inflammation of the myocardium secondary to viral infection. The clinical presentation of viral myocarditis is very heterogeneous and can range from nonspecific symptoms of malaise and fatigue in subclinical disease to a more florid presentation, such as acute cardiogenic shock and sudden cardiac death in severe cases. The accurate and prompt diagnosis of viral myocarditis is very challenging. Endomyocardial biopsy is considered to be the gold standard test to confirm viral myocarditis; however, it is an invasive procedure, and the sensitivity is low when myocardial involvement is focal. Cardiac imaging hence plays an essential role in the noninvasive evaluation of viral myocarditis. The current coronavirus disease 2019 (COVID-19) pandemic has generated considerable interest in the use of imaging in the early detection of severe acute respiratory syndrome coronavirus 2 (SARS-CoV-2)-related myocarditis. This article reviews the role of various cardiac imaging modalities used in the diagnosis and assessment of viral myocarditis, including COVID-19-related myocarditis.

## 1. Introduction

Myocarditis is a disease characterized by inflammation of the myocardial tissue, and it can be either infectious or non-infectious in etiology. Infectious causes include viruses, bacteria, fungi, and protozoa, with viruses being the leading cause of infectious myocarditis [1]. Viral myocarditis is seen in all age groups; however, neonates, children, and immunocompromised individuals are more commonly affected. The clinical presentation of viral myocarditis is heterogeneous and can be acute, subacute, or chronic in nature. The clinical features are nonspecific, with patients presenting with fatigue in subclinical disease to more fulminant disease associated with respiratory failure from acute decompensated heart failure, cardiogenic shock, arrhythmia, and sudden cardiac death in severe or fulminant cases. Occasionally, the pericardium can be involved, leading to myo-pericarditis with patients presenting with pleurisy and pericardial effusion [2].

The incidence of viral myocarditis is not exactly known, which is likely due to the challenges in confirming the diagnosis of viral myocarditis, as the recognized confirmatory test, endomyocardial biopsy (EMB), is infrequently obtained, and there is no noninvasive “gold standard” test. Moreover, the sensitivity of EMB is low and may be falsely negative, especially if myocarditis is focal [3,4]. Many viruses have been documented as causing myocarditis. The most common viral pathogens include coxsackieviruses, parvovirus B19, influenza virus, and, recently, coronaviruses [5].

Overall, an accurate diagnosis of viral myocarditis is dependent on a careful history and physical examination, cardiac biomarkers, electrocardiogram (ECG), and especially noninvasive cardiac imaging [6].

The current coronavirus disease 2019 (COVID-19) pandemic has shown that, in addition to respiratory involvement in the form of acute respiratory distress syndrome (ARDS), the heart can also be affected. Cardiac complications from COVID-19 include acute coronary syndrome (ACS), heart failure, cardiogenic shock, arrhythmias, and myocarditis. Myocarditis after coronavirus infection appears to be the most common cardiac complication, with about 7% mortality [7].

Early diagnosis with noninvasive cardiac imaging and aggressive and prompt treatment can help reduce cardiovascular morbidity and mortality related to COVID-19 infection [8,9]. This review article discusses the role, advantages, limitations, and evidence of various cardiac imaging techniques and modalities in the diagnosis and workup of viral myocarditis, with special focus on COVID-19-related myocarditis.

## 2. Role of Imaging Modalities in Myocarditis

### 2.1. Echocardiography

Transthoracic echocardiography (TTE) is a safe, widely available, and clinically extremely useful cardiac imaging tool, particularly for the initial assessment of viral myocarditis. The American College of Cardiology (ACC), American Heart Association (AHA), and European Society of Cardiology (ESC) Working Group on Myocardial and Pericardial Diseases recommend that all patients with clinically suspected myocarditis should undergo a TTE at initial presentation [10,11,12,13,14]. Poor acoustic windows in patients with obesity or chronic lung diseases is a well-known limitation of echocardiogram and can lead to an inadequate assessment of cardiac function and structure [15]. In general, echocardiographic features of acute myocarditis are subtle, with focal wall motion abnormalities and mildly reduced ejection fraction [16].

Advanced echocardiographic tools, such as three-dimensional (3D) imaging, speckle tracking, contrast echocardiography, and tissue Doppler imaging, can detect subtle abnormalities in ventricular function that can provide clues for the diagnosis of viral myocarditis [17].

#### 2.1.1. Two-Dimensional Transthoracic Echocardiography

Acute viral myocarditis is usually associated with echocardiographic findings of left, right, or biventricular systolic and diastolic dysfunction. Viral infection of the myocardium leads to the infiltration of acute inflammatory cells in myocardial tissue, resulting in interstitial edema with a thickening of the ventricular wall, increased left ventricular (LV) mass index, and reduced ventricular contractility (refer to Figure 1) [18,19,20]. Usually, right ventricular (RV) dysfunction is accompanied by LV dysfunction. RV dysfunction is associated with increased morbidity and mortality and the need for heart transplantation. Pinamonti et al. reviewed echocardiographic studies of 42 patients with biopsy-proven myocarditis. In this study, a total of 23% of the patients had evidence of RV dysfunction, and the presence of RV dysfunction was associated with a worse prognosis [19]. Myocarditis may cause segmental or global dilatation of the LV, focal thickening of the ventricular wall, regional wall motion abnormalities, pericardial effusion, and focal interstitial edema of the myocardium (refer to Table 1). In several cases of acute myocarditis confirmed by biopsy, TTE was reported to be normal, especially in mild cases [20]. The likely explanation for a normal TTE is that either mild myocarditis does not significantly impact the left ventricular function or the changes in the left ventricular function are too subtle to be detected by 2 Dimensional (2D) echocardiography. In addition, echocardiographic findings cannot differentiate viral myocarditis from other forms of cardiomyopathy [16].

#### 2.1.2. Speckle Tracking Echocardiography

Speckle tracking echocardiography (STE) technology has increased accuracy for the diagnosis of LV and RV systolic and diastolic dysfunction as compared to conventional 2D TTE, especially in patients with acute myocarditis. STE, with its capability to differentiate normal contractility from translation motion of the myocardium, can be used to quantify regional contractile function. STE is a time-consuming technique that requires considerable expertise and may not be readily available at most centers [21,22,23].

Strain and strain rate, as measured by STE, have been shown to diagnose early ventricular dysfunction and predict prognosis. Strain imaging can detect subtle LV dysfunction in patients with acute or subacute viral myocarditis, where conventional TTE showed preserved LV function. 

Logstrup et al. studied left ventricular function with conventional echocardiography and STE in 28 patients diagnosed with acute myocarditis based on the Lake Louise criteria. Comparing left ventricular function with 2D echocardiography versus strain imaging, the global longitudinal (−16.2 ± 3.6%), epicardial longitudinal (−14 ± 3%), and endocardial longitudinal (−19.4 ± 3.9%) systolic strains were significantly reduced despite normal left ventricular function on 2D echocardiography. Strain imaging demonstrated a good correlation with the degree of myocardial edema [24]. Other case reports have demonstrated that STE measurements are more sensitive than a 2D TTE in identifying subtle regional wall motion abnormalities and diagnosing acute viral myocarditis [25,26].

#### 2.1.3. Tissue Doppler Imaging

The tissue Doppler imaging (TDI) technique measures the velocity of myocardial motion instead of the velocity of blood flow. TDI is useful in assessing global and regional LV systolic function, LV diastolic function, and left ventricular filling pressures. Tissue Doppler indices tend to be abnormal in patients with acute myocarditis [27]. Urhausen et al. reported a case of myocarditis in a 31-year-old athlete with normal 2D and Doppler echocardiograms, as well as CMR. The TDI did show a net loss of systolic regional wall velocity. The diagnosis of chronic myocarditis was confirmed on EMB [28].

TDI shows promise; however, further research is still required to determine the role of TDI in acute myocarditis.

#### 2.1.4. Contrast Echocardiography

Contrast echocardiography can help in the accurate assessment of left ventricular function and regional wall motion abnormalities and to detect left ventricular thrombi in acute myocarditis [29]. Afonsa et al. [30] described a case of a seventeen-year-old male with viral myopericarditis. A 2D TTE revealed a reduced left ventricular ejection fraction (LVEF) of 35%, asymmetrical thickening of the infero-lateral wall, dyskinesis of the inferolateral wall, and a small pericardial effusion. Using a novel application of echo contrast, attenuated perfusion with delayed contrast replenishment was seen in the inferolateral segments, leading to a strong suspicion of myocarditis. A CMR study in this patient confirmed the observation noted on the echocardiogram. However, the perfusion application of contrast echocardiography is still experimental [30].

#### 2.1.5. Three-Dimensional (3D) Transthoracic Echocardiography

TTE has been shown to be useful for studying and understanding complicated cardiac anatomies and hemodynamics. Its role in the diagnosis of acute viral myocarditis is not yet clear [31]. Thuny et al. reported the utility of a 3D TTE in a 43-year-old male with acute myocarditis. On a 2D TTE, the patient had LV hypokinesis with impaired LV contractility and biventricular thromboses, which were better visualized using a 3D TTE [32].

Overall, the sensitivity and specificity of these newer and more advanced echocardiographic techniques in diagnosing viral myocarditis are currently unknown and need to be studied further against cardiac magnetic resonance imaging (CMR) and EMB [17].

### 2.2. Cardiac Computed Tomography

The clinical presentation of acute viral myocarditis is extremely variable, and multi-detector computed tomography (MDCT) can help in ruling out other conditions that may mimic viral myocarditis, such as ACS, aortic dissection, acute pulmonary embolism (PE), congestive heart failure, and pneumonia [33]. Acute myocarditis is associated with increased permeability of the inflamed myocardium, resulting in an increased uptake and accumulation of radiographic contrast agents. Iodinated contrast agents used with MDCT share common pharmacokinetics with the gadolinium-based contrast agents (GBCA) used with CMR. Hence, late myocardial enhancement imaging techniques used with CMR in the evaluation of acute myocarditis are also applicable to MDCT. The typical findings in acute myocarditis on MDCT are therefore similar to those of CMR, which are seen as delayed midwall or subepicardial myocardial enhancement on iodine contrast. Cardiac CT in addition can help differentiate between ACS and myocarditis by demonstrating the absence of significant coronary artery disease during the same examination. In addition, cardiac CT can also detect global and regional wall motion abnormalities of the left ventricle [33,34].

Even though CMR is considered the primary imaging technique for the diagnosis of acute myocarditis, MDCT has a few advantages over CMR [35]. MDCT is readily available and more accessible, and it has a shorter scanning time as compared to CMR. Cardiac CT imaging can provide coronary artery examination and rule out ACS in suspected patients. Moreover, it may be a reasonable alternative when CMR is not an option (e.g., in patients with metallic implants or claustrophobia) [36,37]. Bouleti et al. studied 20 patients admitted with chest pain and elevated troponin I, who were diagnosed with acute myocarditis by CMR. These patients then had spectral cardiac CT with late iodine enhancement (refer to Figure 2). Spectral CT showed an overall accuracy of 95% in the diagnosis of acute myocarditis compared to CMR [38]. From the limited data available, cardiac CT appears to have some useful application in viral myocarditis; however, the role of cardiac CT in acute viral myocarditis is still not well defined.

### 2.3. Cardiac Magnetic Resonance Imaging

The histopathology of myocarditis includes inflammatory response, edema, and endothelial dysfunction, followed by myocyte necrosis and fibrosis [39]. CMR has become the leading cardiac imaging modality for tissue characterization; it has excellent spatial resolution, and acceptable interobserver variability and quantitative accuracy [40,41,42,43]. CMR is able to detect myocardial edema, hyperemia, necrosis, and fibrosis. Thus, CMR is the first choice for the assessment of myocarditis, as well as for monitoring disease activity while being treated [39,44]. The Lake Louise Consensus Group proposed a standard CMR protocol to identify the tissue targets in myocarditis. These diagnostic targets include edema and hyperemia, as well as necrosis and fibrosis. Any two out of the three Lake Louise criteria (LLC) establish a positive imaging diagnosis of acute myocarditis with a diagnostic accuracy of 78%, a sensitivity of 67%, and a specificity of 91% [40,45,46,47]. The presence of regional or global systolic LV dysfunction and pericardial effusion are considered supportive criteria [39,47].

In CMR, a T2-weighted imaging sequence is used to detect myocardial edema, conventionally obtained using black-blood spin-echo techniques. Edematous myocardium causes prolonged T2 decay times, which can be seen as hyperintense signals on T2-weighted images [46,48,49]. T2-weighted imaging can be evaluated using a semi-quantitative method by comparing the signal intensity (SI) of myocardium to skeletal muscle as a reference region of interest (ROI). Some studies showed that a myocardium-to-skeletal muscle SI ratio of more than 1.9 on T2-weighted CMR imaging has a sensitivity of 84%, a specificity of 74%, and an overall accuracy of 79% to detect significant myocarditis [46,50]. A limitation of this technique is the reference ROI; if the reference skeletal muscle is inflamed as seen in those with systemic inflammatory conditions, one could obtain false-negative results [46,47,51].

Another CMR technique, i.e., early T1-weighted enhanced sequence, is acquired one minute after administering gadolinium. This technique relies on the detection of myocarditis-related hyperemia, and the early gadolinium enhancement (EGE) sequence shows affected areas as hyperintense signals [48,49]. Semi-quantitative methods can be used by comparing the myocardial-to-skeletal muscle SI ratio before and after giving GBCA. The EGE sequence has many limitations due to image quality inconsistency and variability in SI by using different CMR systems. Data have shown that removing this EGE criterion from the original LLC does not remarkably affect the diagnostic accuracy of myocarditis [48,49,52]. Although EGE imaging is still being used in some experienced centers, it is no longer needed as a diagnostic criterion in the revised LLC [46,47]. 

Late gadolinium enhancement (LGE) imaging taken ten minutes after administering GBCA shows the accumulation of gadolinium in areas of necrosis and fibrosis. Myocyte cell membrane destruction results in the passive diffusion of gadolinium from the extracellular space into the intracellular space. When delayed images are taken, the areas of inflammation appear as a hyperintense signal on T1-weighted imaging compared to normal myocardium. The patterns of LGE hyperintense signals in myocarditis are heterogeneous. The most common patterns are patchy, non-adjacent distributions seen mostly in the mid-myocardial and/or subepicardial areas in the septal or lateral walls [46,48,49,53]. This pattern typically helps to differentiate myocarditis from ischemic cause, which is associated with subendocardial enhancement pattern. Occasionally, transmural involvement of the myocardium can be seen with extensive myocarditis. LGE imaging alone is limited given its inability to differentiate active from chronic myocarditis [46,54]. When LGE images, which detect irreversible changes of necrosis and fibrosis, are compared to T2-weighted images, which detect early changes seen in myocarditis, such as edema, the acuity of myocarditis can be estimated [49,55,56].

Novel CMR techniques, particularly T1 and T2 mapping, as well as extracellular volume (ECV) quantification, have appeared to be accurate methods to characterize myocardial edema [40]. Body tissues have predictable T1 and T2 relaxation times. Any physiologic or pathologic change in tissue structure is noted as deviation from the normal T1 and T2 relaxation times. In acute myocarditis, edema causes significant prolongation of myocardial T1 and T2 relaxation times. These mapping techniques give quantitative data of tissue magnetic properties without subjective limitations of visual assessment of T2-weighted imaging and SI in EGE imaging [40,46]. The most commonly used methods for T2 mapping are gradient and spin echo, and for native T1 and ECV mapping, they are inversion recovery and saturation recovery sequences [46,56,57,58,59,60,61]. T2 mapping is a very reliable technique to detect myocardial edema without the disadvantages of qualitative T2-weighted imaging (see Figure 3). Since native T1 is sensitive to intra- and extra-cellular free water content, the T1 relaxation time increases in acute inflammation and hyperemia [62,63]. Native T1 mapping and ECV mapping are applicable tools for fibrosis evaluation [38]. Inflammation can be seen directly on native T1 and T2 mapping without the need to use contrast agents. When GBCA is used, combined pre-contrast and post-contrast T1 mapping can be used to quantify ECV in acute and chronic myocarditis [40,64]. However, ECV mapping is a more demanding technique, as it needs acquisition of T1 maps before and after administering GBCA and hematocrit adjustment. When compared to LGE, which detects focal fibrosis, ECV mapping can be complementary to LGE given its ability to detect milder and more diffuse myocardial fibrosis [38,47,62,65]. Other advantages of mapping techniques include the lack of need for a reference ROI and the shorter breath-holding time required. T1 and T2 mapping have demonstrated good sensitivity to identify myocardial inflammation [46,47,66]. A meta-analysis by Kotanidis et al. showed higher sensitivity of T1, T2, and ECV mapping compared to the standard CMR techniques (refer to Table 2) [67].

Another CMR novel technique is CMR image-derived myocardial strain analysis, which uses different methods to quantitatively assess myocardial deformation. Myocardial strain can help detect subtle systolic or diastolic dysfunction, which cannot be seen on routine imaging. Its diagnostic value becomes more evident when combined with T2 mapping and LGE [40,68].

Given the remarkable evolution in CMR technology, including quantitative T1 and T2 mapping techniques, the LLC were revised in 2018 to include parametric mapping [46]. In clinically suspected myocarditis, both T1 and T2 criteria must be present according to the 2018 revised LLC (refer to Figure 1) [46,67]. The fact that the 2018 revised LLC is a GBCA-free protocol, it has given it a great advantage over the original LLC, especially when CMR is considered in patients who cannot tolerate GBCA, such as those with an allergy to GBCA, pregnant patients, and those with end-stage renal insufficiency [38].

A study was conducted by Luetkens and colleagues to compare the 2018 revised LLC to the original LLC in diagnosing acute myocarditis. The study revealed that the 2018 revised LLC has a significantly higher sensitivity compared to the original LLC (87.5% vs. 72.5%; *p*-value = 0.031), with no difference in specificity (96.2% vs. 96.2%; *p*-value = 0.999). It concluded that the 2018 revised LLC has a better diagnostic performance of CMR in acute myocarditis [38,46,67,69,70].

### 2.4. Nuclear Scintigraphic Imaging

Myocardial scintigraphy with inflammation-sensitive radioisotopes has been used to diagnose acute myocarditis [71]. The isotopes that have been used in the workup of acute myocarditis include gallium-67 (Ga-67), indium-111 (In-111) monoclonal antimyosin antibody, and technetium-99m (Tc-99m)-labeled methoxy-isobutyl isonitrile (MIBI) single-photon emission computed tomography (SPECT), and technetium-99m depreotide [71].

Lymphocyte labeling techniques using gallium-67 scintigraphy can detect areas of inflammation in patients with myocarditis and can potentially differentiate myocarditis from acute myocardial infarction [72,73,74]. O’Connell et al. [73] reported a case series comparing Ga-67 scintigraphy imaging with EMB for the diagnosis of myocarditis in patients with dilated cardiomyopathy. Five out of six cases of EMB-proven myocarditis showed a dense uptake of Ga-67, suggesting that screening Ga-67 scintigraphy can increase the yield of myocardial biopsy.

Antimyosin, a monoclonal antibody against cardiac myosin, radiolabeled with In-111, has been used with scintigraphy in the diagnostic workup of acute myocarditis [75,76,77,78]. Martin et al. [79] studied antimyosin scintigraphy in 40 pediatric patients with clinically suspected myocarditis. In their observation, the uptake of antimyosin antibodies by the myocardium correlated well to the histological/pathological diagnosis of myocarditis, and persistent antimyosin uptake was associated with increased morbidity [79]. Similar observations were reported by Kuhl et al. [80]; in a study of 65 patients with clinically suspected myocarditis, monoclonal antimyosin antibody radiolabeled with In-111 uptake correlated well with the histoimmunopathological findings from EMB. In-111 antimyosin scintigraphy displayed excellent specificity but poor sensitivity for the detection of acute myocarditis as compared to EMB [80].

Myocardial scintigraphy with radiolabeled Tc-99m MIBI is commonly used to determine myocardial perfusion. The normal uptake and clearance of Tc-99m MIBI by myocardial cells depend on their viability and myocardial cell membrane integrity. In acute myocarditis, myocardial inflammation and necrosis result in abnormal and reduced scan uptake [16]. Sun et al. investigated 46 children with Coxsackie viral myocarditis using a Tc-99m MIBI myocardial perfusion scan, which showed areas of hypoperfusion in all patients [81]. Currently, no myocardial scintigraphy studies are available in patients with COVID-19 myocarditis.

In the recent era, CMR has become widely available and provides several advantages over nuclear imaging, and it has diminished the use of nuclear scintigraphic imaging in acute myocarditis. Some advantages of CMR over nuclear imaging include better spatial resolution, no radiation exposure, and a better correlation with histopathology [82,83,84,85,86,87].

### 2.5. Combined Positron Emission Tomography and Computed Tomography (PET-CT)

Combined 18F-fluorodeoxyglucose (18F-FDG) positron emission tomography (PET) with CT has been investigated as a tool to diagnose myocardial inflammation. One of the potential advantages of PET-CT over CMR is that it can quantify the degree of inflammation, leading to close monitoring of the disease course and the response to anti-inflammatory and immunosuppressive therapies. PET-CT can be considered as an alternative study in patients with contraindications to CMR [88]. Researchers have looked at simultaneous/hybrid cardiac PET-CT/CMR imaging and found that they complement each other in the assessment of myocarditis compared to either approach alone [89]. A number of case reports have demonstrated the use of PET-CT in conditions resulting in myocardial inflammation, such as cardiac sarcoidosis, viral myocarditis, giant cell myocarditis, and post-infarction myocarditis [90,91,92,93,94,95,96]. One prospective study investigated the use of PET-CT as compared to CMR with LGE in 65 patients with suspected myocarditis. It showed that the sensitivity and specificity of PET-CT was 74% and 97%, respectively, with an overall accuracy of 87% as compared to CMR [97]. A clinical trial that is underway plans to assess patients with clinically suspected myocarditis using TTE, nuclear SPECT imaging, and 18F-FDG PET-CT. Right ventricular biopsies will be performed and will be the gold standard for analysis. The trial is designed to look at the sensitivity and specificity of 18F-FDG PET-CT imaging in diagnosing acute myocarditis [98].

## 3. The Role of Imaging in Determining Prognosis in Acute Myocarditis

Most cases of acute myocarditis tend to regress over time, leaving no or only mild functional damage. However, acute fulminant myocarditis is associated with a much worse outcome and more residual damage [12,20].

A retrospective, single-center, observational study of 112 patients diagnosed with acute myocarditis by CMR, studied the primary endpoint of major adverse cardiovascular events (MACE), which included all-cause mortality, cardiac mortality, the recurrence of myocarditis, heart failure, and sustained ventricular tachycardia. The MACE rate was significantly higher in those with extensive LGE on CMR, as defined as those with LGE of more than 17 g, compared to those with LGE less than 17 g (MACE rate of 17% vs. 4%; *p*-value = 0.005). Moreover, those who initially presented with an infarct-like pattern of myocarditis, defined as those who presented with chest pain, ST elevation on ECG, and elevated cardiac enzymes (troponin), were noted to have an increased risk of MACE recurrence and particularly sustained ventricular tachycardia. During a median follow up of 16 months, those with initial symptoms that correlated with New York Heart Association (NYHA) class III or IV and LGE of more than 17 g on CMR were independent predictors of MACE occurrence after acute myocarditis [99].

Based on two meta-analyses, CMR might be a helpful tool in determining the long-term prognosis of acute myocarditis. They were able to show that LGE and LVEF are strong predictors of MACE, defined as all-cause mortality, cardiac mortality, the recurrence of myocarditis, heart failure, and sustained ventricular tachycardia. However, using cut-off values of LGE (17 g or 13% of myocardial mass) is not a validated approach to be applied in routine clinical practice [99,100]. In a study of 374 patients with acute myocarditis and normal LVEF, the location of LGE determined prognosis; LGE seen in the midwall of the anteroseptal segment is associated with a worse prognosis than LGE seen in other segments [101]. There is no current evidence to support the use of the novel CMR techniques (T2, T1, and ECV mapping) to determine the prognosis of those with acute myocarditis [39].

## 4. COVID-19-Related Myocarditis

COVID-19 disease is caused by severe acute respiratory syndrome coronavirus 2 (SARS-CoV-2). This novel virus came to the forefront of global attention in December 2019 after it was found to cause ARDS in patients from Hubei province in China [7]. With the spread of the virus to countries all over the world and an increasing number of cases, the World Health Organization declared COVID-19 a global pandemic on March, eleventh 2020 [102].

COVID-19 has multisystem involvement, including the cardiovascular system. Acute myocarditis is a recognized complication of COVID-19, although the exact mechanism is not well known. Myocardial injury and ischemic necrosis can be caused be a number of different mechanisms, including direct involvement by the SARS-CoV-2 virus and indirect involvement with myocardial damage caused by coronary thrombosis, coronary plaque rupture, cytokine storm with systemic inflammation, hypoxia, relative ischemia from a supply/demand mismatch, and electrolyte derangements [103]. Only a small number of cases suggesting direct viral involvement of cardiac myocytes resulted in viral myocarditis [104,105].

The clinical presentation of COVID-19 cardiac disease can be heterogeneous, ranging from an asymptomatic cardiac biomarker elevation to severe disease in the form of acute cardiogenic shock and cardiac arrest [105]. Clinically elevated cardiac biomarkers, i.e., serum troponin and BNP, abnormal EKG, and new left ventricular dysfunction, can raise the suspicion of COVID-19 myocarditis; however, an endomyocardial biopsy is the only diagnostic modality for COVID-19 myocarditis.

## 5. Imaging in COVID-19

In the acute setting of acute COVID-19 pneumonia, diagnosing acute myocarditis can be very challenging. An abnormal EKG and elevated cardiac biomarkers can raise the suspicion of COVID-19 myocarditis.

The causal relationship of myocarditis with COVID-19 can be very difficult to determine, especially in the setting of ARDS while dealing with an unstable and ventilated population [106,107,108]. Advanced imaging might be necessary to support the diagnosis of myocarditis and can perhaps help to differentiate between cardiovascular and pulmonary causes of such an overlapping presentation of COVID-19 infection [109]. A pandemic of a highly contagious pathogen poses an important ethical dilemma; i.e., one has to be judicious while ordering cardiac imaging in patients with a highly contagious infection to prevent the risk of transmission to the operators and staff members. Reports have shown that echocardiography poses a higher risk of transmitting COVID-19 when compared to CMR. It is critical for the staff to practice and adhere to protective measures, including the proper use of personal protective equipment (PPE) and the disinfecting of equipment (ultrasound probe and scanners) and imaging rooms [109,110,111,112]. It is of paramount importance to minimize the time of exposure by performing only a focused echocardiogram rather than a full exam in patients under investigation (PUI) or confirmed COVID-19 patients [113,114].

Despite the clues of myocarditis on imaging, EMB is considered to be the gold standard for the diagnosis of myocarditis. The findings of EMB on the histopathology of COVID-19-related myocarditis have been described as cellular infiltrates with necrotic areas. EMB and autopsy specimen findings do not show any specific or reproducible pattern of COVID-19 myocarditis. In addition, some case reports early in the pandemic did not show any direct viral involvement of the myocardium on EMB or even autopsy. In some reports, the viral genome was detected on tissue specimens [103]. The current ESC guidelines do not recommend cardiac biopsy for COVID-19 patients with suspected myocarditis [112]. The limited role of EMB makes noninvasive cardiac imaging modalities more vital to diagnose COVID-19-related myocarditis.

### 5.1. Transthoracic Echocardiography in COVID-19

Given its low cost, bedside availability, and portability, TTE is considered to be the initial imaging modality in suspected myocarditis in COVID-19-related ARDS. Many hospitals have dedicated ultrasound machines to be used only in COVID-19 units [110,112,113,114,115,116]. The ESC Working Group on Myocardial and Pericardial Diseases recommended a TTE as part of the initial workup for all COVID-19 patients with suspected myocarditis [12,63,111]. Left ventricular or biventricular dysfunction, altered ventricular global strain, myocardial edema, LV thrombus, and pericardial effusion can be seen in myocarditis related to COVID-19 but are nonspecific findings (see Figure 4) [105,117].

A systematic review by Rathore et al. examined data from case reports and a case series of 42 patients with COVID-19-related myocarditis. TTE was performed in 35 patients, 74% of whom showed low LVEF with a mean of 37%. Other important echocardiographic features included LV hypokinesis (37.2%) and pericardial effusion (26% of patients) [118].

In a large series of 218 COVID-19 patients with no underlying cardiac disease, speckle tracking echocardiography demonstrated abnormal strain, i.e., reduced global longitudinal shortening in 83% of patients, while only 22% of patients were noticed to have left ventricular dysfunction on 2D echocardiography [119]. A reduced GLS was more commonly seen in critically ill patients (98% vs. 78.3%, *p* < 0.001). The average GLS was −13.7% ± 3.4% vs. −17.4% ± 3.2%, *p* < 0.001 in the critically ill patients as compared to noncritical patients. The distribution of the strain was noticed more in the subendocardial regions, which is a typical pattern of myocarditis. The changes in GLS correlated significantly to clinical and inflammatory markers, such as pulse oxygen saturation, high-sensitive C-reactive protein, and inflammatory cytokines, especially in sicker patients [119]. Another retrospective study from Croft et al. studied LV GLS in 58 non-consecutive patients hospitalized with COVID-19 infection. The mean LV ejection fraction (LVEF) and LV GLS was 52.1 and −12.9 ± 4.0%, respectively. In the 30 patients with preserved LVEF (>50%), LV GLS was also reduced (−15.7 ± 2.8%) compared to the healthy population. Data from these studies indicate that acute myocardial injury may be subtle in patients with COVID-19 and that strain imaging can be useful for the identification of occult myocardial injury [120].

In a case report by Trogen et al. of a 17-year-old patient with SARS-CoV-2-related acute myocarditis, confirmed by echocardiography and cardiac magnetic resonance imaging, tissue Doppler abnormalities and abnormal strain were present even after one week of discharge [121]. These observations suggest that some of the sequelae related to COVID-19 myocarditis can be persistent. More studies are required to understand if these changes will reverse after some time or whether they will lead to permanent dysfunction.

Advanced echo techniques, such as strain imaging and tissue Doppler imaging, can be used as gatekeeper tests to identify patients who can benefit from CMR and EMB for the diagnosis and follow-up of COVID-19 myocarditis [122,123].

### 5.2. Cardiac Computerized Tomography in COVID-19

The presentation of COVID-19 has overlapping cardiac and respiratory involvement. Moreover, acute PE is very common in COVID-19 pneumonia and can also present with an abnormal EKG and elevated biomarkers; however, these patients tend to have severe hypoxemia and tachycardia disproportionate to the degree of pneumonia. Cardiac CTA can be a one-stop shop for the diagnosis of acute PE, excluding coronary artery disease [124,125], as well as for the assessment of the severity of pulmonary disease.

The utilization of cardiac CTA in COVID-19 patients can prevent unnecessary coronary angiograms and the exposure of cardiac catheterization laboratory personnel to COVID-19 [126]. A delayed post-iodine contrast CT scan be useful for tissue characterization in COVID-19-related myocarditis [127]. One health care system in New York reported their experience of using CCTA to evaluate patients presenting with acute chest pain during the COVID-19 pandemic. They detected two cases of acute peri-myocarditis among ten confirmed COVID-19-positive patients using CCTA [128].

CT is limited in identifying myocardial edema. Due to its inability to null the signal from normal myocardial tissue, CT with delayed enhancement is still inferior to CMR [129,130,131]. The ESC, the European Association of Cardiovascular Imaging (EACVI), and the Society of Cardiovascular Computed Tomography (SCCT) have recommended the use of CCTA in COVID-19 patients with acute chest pain, especially if it is expected to impact their management [112,132,133]. Further studies about CCTA are still needed to provide more information on its utility in managing COVID-19 patients with cardiac involvement.

### 5.3. Cardiac Magnetic Resonance Imaging in COVID-19

CMR is considered the gold standard, noninvasive diagnostic tool for suspected myocarditis [36,134]. Key findings in COVID-19 myocarditis include myocardial edema, myocardial necrosis, LGE, and RV dysfunction [135]. Panchal et al. reported that, from their experience, CMR demonstrated more diffuse myocardial involvement in COVID-19-related myocarditis as compared to non-COVID-19 myocarditis [136].

In a large cohort of 100 German patients who recovered from COVID-19, abnormal myocardial findings were noted in 78% of them, which included myocardial edema, LGE, and pericardial enhancement [137]. In CMR of 26 college athletes who recovered from COVID, 15% of them (*n* = 4) met the updated LLC criteria for myocarditis, i.e., the presence of myocardial edema and myocardial injury by LGE. An additional 46% of the athletes were found to have late gadolinium enhancement alone [138]. On the screening of 1597 athletes who recovered from COVID-19, 2.3% of them were found to have findings consistent with COVID-19 myocarditis (9 had clinical symptoms and 27 had subclinical symptoms of myocarditis). These observations support the fact that the use of CMR can improve the detection of COVID-19 myocarditis, particularly in individuals with subclinical symptoms [139].

A midterm follow-up of recovered COVID-19 patients showed that LGE was present in 30% of the patient population. These patients tend to have a lower LV circumferential strain and abnormal RV strain parameters [140]. The presence of an abnormal myocardial pattern after complete recovery from COVID-19 suggests that there might be long-term cardiac sequelae of COVID-19 infection, which are largely unknown.

CMR has its own challenges, which include scant availability; cost; exam length; and patient-related challenges, such as arrhythmia, inability to hold breath, claustrophobia, implanted metallic devices, and contrast allergy [63,68,141]. CMR testing is very challenging in COVID-19 patients who are intubated. Experts recommend performing a focused CMR using specific sequences instead of a comprehensive exam to reduce CMR time. This can be achieved through using a modern magnet. A short protocol CMR with the acquisition of T2 mapping and steady-state free precession (SSFP) are enough to evaluate regional wall motion abnormalities, systolic function, chambers size and volume, and the presence of edema. Once patients are more stable and no longer infectious, a complete CMR protocol with LGE can be carried out to assess for fibrosis [63,142].

### 5.4. Nuclear Imaging in COVID-19

Nuclear cardiology imaging techniques require a long acquisition time and special protocols, which increase the risk of contracting COVID-19. In COVID-19 patients, the use of nuclear cardiology imaging modalities should be limited to those with a clinical suspicion of cardiac involvement when other imaging modalities cannot be used or are contraindicated. Examples include those with prosthetic valves or intracardiac devices [109,143]. The role of PET-CT in COVID-19-associated myocarditis is not clear. There are some reports on the use of PET-CT in COVID-19 patients that look specifically at acute respiratory disease where lung involvement, seen as acute lung inflammation, is detected on PET-CT. Given the fact that PET-CT can detect acute viral myocarditis, it is possible to apply that to COVID-19-related myocarditis [144,145].

The role of combined cardiac CT and 18F-FDG PET-CT in COVID-19 infections is currently being evaluated against the gold standard CMR. The primary endpoint of the study is the proportion of COVID-19 subjects with cardiovascular injury within two weeks of admission [146]. More data and studies are still needed to clarify the role and applicability of 18F-FDG PET-CT scanning in COVID-19-related myocarditis.

## 6. Conclusions

Cardiovascular disease secondary to SARS-CoV-2 includes acute myocarditis and is associated with significant morbidity and mortality. An accurate diagnosis of acute viral myocarditis is very challenging with the heterogeneous presentation of the disease. Despite its own disadvantages, EMB is considered to be the gold standard diagnostic test. Myocardial involvement in viral myocarditis is focal and patchy, which explains the low sensitivity and diagnostic accuracy of EMB. Therefore, noninvasive cardiac imaging is an essential part of the workup of acute viral myocarditis. CMR is considered to be the most useful noninvasive test to detect myocarditis and can give detailed information in terms of myocardial structure and function. Noninvasive imaging modalities, such as TTE, along with strain imaging and tissue Doppler analysis, can help to identify myocardial abnormalities, which can be subsequently confirmed with CMR. At present, the role of CT in the diagnosis of COVID-19 myocarditis is not known. 

Nuclear imaging can be helpful to identify myocardial scar; however, the role of nuclear imaging will remain limited in the era of CMR. The proper use of PPE and meticulous decontamination techniques is essential in reducing the risk of viral transmission of this highly contagious virus. Further research is needed to improve our understanding, approach, management, and follow-up of COVID-19-related myocarditis.

## Data Availability

Not applicable.

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
