# Peer review of "A Review of the Role of Imaging Modalities in the Evaluation of Viral Myocarditis with a Special Focus on COVID-19-Related Myocarditis"

_diagnostics, 2022, doi:10.3390/diagnostics12020549_

Round 1

Reviewer 1 Report

Well documented article. Too long and with unnecessary repetitions. I suggest to change the structure and when you speak about one imaging method complete with special considerations - if they are really present - about COVID induced particularities.

Sure all of us are fascinated by COVID pandemic but myocarditis remain....myocarditis with their well known difficulties in diagnosis and establishing prognosis. What's about COVID and ischemic myocardial necrosis?

I looking for something more exciting if really we can find something very unusual in COVID induced myocarditis imaging

Author Response

Reviewer 1 comments and suggestions

Response

Too long and with unnecessary repetitions

As suggested by the reviewer, we have reviewed and edited the manuscript. The repetitive parts of the manuscript have been deleted. 

Change structure and when you speak about one imaging method complete with special considerations - if they are really present - about COVID induced particularities.

We appreciate the recommendation by the reviewer, the recommendations were carefully considered. We feel COVID-19 related finding requires a separate discussion and would prefer to keep the current format. 

What's about COVID and ischemic myocardial necrosis?

This is a very important comment by the reviewer, ischemic myocardial necrosis is more common. We have updated the manuscript and added disc

ussion on COVID and ischemic myocardial necrosis in section 4 and 5.3 ( COVID-19 related myocarditis )

something very unusual in COVID induced myocarditis imaging

Added under CMR and COVID section  : CMR showed more diffuse involvement of myocardium in COVID-19 induced myocarditis compared to non-COVID myocarditis  

(Panchal A, Kyvernitakis A, Mikolich JR, Biederman RWW. Contemporary use of cardiac imaging for COVID-19 patients: a three center experience defining a potential role for cardiac MRI. Int J Cardiovasc Imaging. 2021;37(5):1721-1733. 

Reviewer 2 Report

I read the article entitled "A Review of the Role of Imaging Modalities in the Evaluation of Viral Myocarditis with a Special Focus on COVID-19 Related Myocarditis". Here the authors provided a nice and comprehensive overview on the non-invasive diagnosis of myocarditis. The paper is well written and sufficiently covers the main areas of interest. I have the following comments:

1) CT paragraph: based on the amount of published literature as well as indication from societal guidelines (eg ESC guidelines on NSTEMI), I suggest to report more carefully the potential role of cardiac CT in evaluating tissue composition in myocarditis. Indeed, to this end, the amount of evidence available for CMR is outweighing, making it the modality of choice in the evaluation of patients with clinically suspected myocarditis/MINOCA.

2)  chart 1: I think it could be deleted, since newer criteria (correctly reported in chart 2) have been established

3) LGE acquisition: need to wait at least 10 minutes (as per consensus document https://doi.org/10.1186/s12968-020-00607-1)

4) CMR prognostic factors: it might be worth noting that LGE location could be associated with prognosis in myocarditis (10.1016/j.jacc.2017.08.044)

5) A further aspect that could be discussed is acute myocarditis in the absence of any LGE. In this cases, common as in example in immunecheck-point inhibitor myocarditis (10.1093/eurheartj/ehaa051,  10.1016/j.jacc.2021.01.050) or COVID-19 myocarditis too (10.1093/ehjci/jeaa414), the use of mapping techniques as well as revised Lake Louise criteria might be essential for diagnosis 

6) Few more images coming from different CV imaging modalities could be useful to enhance the educational value of the manuscript

Author Response

Reviewer 2 comments and suggestions

Response

Report more carefully the role of cardiac CT in evaluating tissue composition in myocarditis

We appreciate the recommendation. 

We have added in section  2.2. : Cardiac computed tomography  

Delete chart 1.  As per reviewer since many of the newer criteria as per chart 2 have been mentioned already

Deleted chart 1 as suggested. 

LGE acquisition (need to wait 10 min) as per consensus document

We have made the recommended changes to the manuscript

Discuss that LGE location could be associated with prognosis in myocarditis

We have edited the manuscript by adding the role of LGE on prognosis. The ITAMY study was added to the discussion.

374 patients with acute myocarditis and normal LVEF, the location of LGE determined the prognosis; LGE seen in the mid-wall of anteroseptal segment is associated with worse prognosis compared to those with LGE seen in other segments [Aquaro GD, Perfetti M, Camastra G, et al. Cardiac MR With Late Gadolinium Enhancement in Acute Myocarditis With Preserved Systolic Function: ITAMY Study. J Am Coll Cardiol. 2017;70(16):1977-1987. doi:10.1016/j.jacc.2017.08.044]

A further aspect that could be discussed is acute myocarditis in the absence of any LGE. In

this cases. common as in example in immunecheck-point innibitor mvocarditis

(10.1093/eurhearti/ehaa051, 10.1016/.jacc.2021.01.050) or COVID-19 myocarditis too

(10.1093/ehjciljeaa414), the use of mapping techniques as well as revised Lake Louise criteria

might be essential for diagnosis

We have compared the novel CMR techniques to LGE  in section 2.3. CMR. 

Suggest a few more images from difference CV imaging modalities

Added an additional image ( figure 1 ) We will work on providing more images.

Round 2

Reviewer 1 Report

Paper was improved. 

But I wish that differential diagnosis of CMR between inflammation and ischemia was discussed more. 

As we already know in patients with COVID, myocardial ischemia is a common manifestation. When we talk about diagnosis, we must also take into account the differential diagnosis, not only the positive one.

So please have some extra considerations here. You don't say anything about coronary angiography - is it  useless in a positive and differential diagnosis? Morever in extensive myocarditis?

Reviewer 2 Report

Thank you for providing appropriate answers to the Reviewers' indications. I have no further comment.

Author Response

Reviewer 2 had previously suggested adding more images ; we have added more images to the manuscript. 

Thank you